# URSA: The Universal Research and Scientific Agent

## Abstract

Large language models (LLMs) have moved far beyond their initial form as simple chatbots, now carrying out complex reasoning, planning, writing, coding, and research tasks. These skills overlap significantly with those that human scientists use day-to-day to solve complex problems that drive the cutting edge of research. Using LLMs in "agentic" AI has the potential to revolutionize modern science and remove bottlenecks to progress. In this work, we present URSA, a scientific agent ecosystem for accelerating research tasks. URSA consists of a set of modular agents and tools, including coupling to advanced physics simulation codes, that can be combined to address scientific problems of varied complexity and impact. This work highlights the architecture of URSA, as well as examples that highlight the potential of the system.

## 1 Introduction

The promise of AI for accelerating science has quickly turned from a far-off vision to a near-term reality for advancing cutting-edge research. Emergent reasoning and planning capabilities in large language models (LLM) have opened new avenues for automating complex science and engineering tasks and eliminating human-driven bottlenecks in the research process. As an example, consider scientific domains like inertial confinement fusion (ICF) and materials modeling. These models rely on physics simulations to explore hypothesis spaces and guide experimental research. Unfortunately, these high-fidelity simulations can take hours or even days on some of the largest supercomputers, often delaying scientific discoveries by months or longer. A major contributor to these inefficiencies is the prevalence of unproductive simulations that fail to advance knowledge, presenting an opportunity for AI to accelerate progress through better identification of simulations to run.

Recent progress in large-scale foundation models and autonomous "agentic" tool use (e.g., code-generating assistants and planner-executor architectures) suggests a path toward AI systems that can process and reason over large amounts of data to decide what to simulate and adapt hypotheses on the fly. However, most demonstrations to date target internet-scale tasks such as software debugging or web search, leaving open the question of how an agent can combine successes in reasoning and coding tasks with high performance scientific computing for high-impact, high-consequence scientific applications.

In this work we present the URSA agentic workflow for accelerating scientific efforts developed at Los Alamos National Laboratory for use by the scientific community. URSA uses a set of modular, composable agents coupled with tool use to hypothesize, plan, and execute research tasks to supplement domain expert expertise in accelerating research outcomes. URSA uses agents dedicated to planning, hypothesizing, researching, and executing computational tasks. Some agents can also leverage advanced scientific simulation such as trusted radiation-hydrodynamics models for ICF simulation.

### 1.1 Contributions

1. **Agentic architecture for scientific tool use:** We introduce URSA which combines large-scale language-model planning, autonomous research, and LLM-driven design optimization.

2. **Composable agents for varying complexity:** Our approach builds on the recent success of Sakana Lu et al. (2024) and similar workflows, introducing an architecture to generalize prescribed, linear processes by incorporating structures that support loops and other feedback mechanisms.

3. **Demonstration:** We present a series of experiments to demonstrate the capability of URSA on increasingly complex problems.

4. **Leveraging physics simulation for design automation** We present results for a key component of simulation-based scientific discovery–utilizing computational models to identify promising designs. We show that URSA outperforms standard methods (Bayesian optimization) for a design optimization task utilizing radiation hydrodynamics simulation.

Our study charts a concrete path toward AI systems that actively *does* science, while illuminating gaps that remain in the autonomous use of agents for critical science applications.

## 2 RELATED WORK

The literature on agentic AI is broad. Two recent surveys, Gridach et al. (2025) and Ren et al. (2025), provide a good overview of recent progress. Both provide a useful categorization of the components of such a system. Gridach et al. (2025) divides the agentic discovery process into ideation; experimental design and execution; data analysis and interpretation; and paper writing and dissemination. They also examine implementation and application datasets. Ren et al. (2025) breaks out the key components of a system in a planner, memory, and tool sets for execution. They further break these down and cover the literature on each, including scientific application domains. Similarly, Zhou et al. (Submitted) reviews several of the top approaches in the context of scientific AI agents.

The Sakana AI Scientist papers Lu et al. (2024); Yamada et al. (2025) are influential exemplars of these end-to-end approaches. This work is the most closely related to what we present here. These represent an attempt to build an end-to-end automated approach to machine learning research. The system generates ideas, reviews them for novelty, and scores them. The selected idea is implemented along with numerical experiments. Finally, the idea is written up and reviewed. Version 2 expanded the system's capability by using tree-search for the experimentation and a vision-language model to improve the figures in the written papers. Both versions use base models without additional fine-tuning. Our approach builds on these ideas by developing agents that support less regimented workflows, such as loops that allow ideas to be approved by feedback.

The Aviary system Narayanan et al. (2024) also represents a complete approach, but focuses on training complete systems of agents with a reinforcement learning approach to fine-tuning LLMs. They introduce the concept of a language decision process and provide two approaches to training. They also consider tool use and decision-making to support it. They find that small LLMs can be trained to match the performance of larger frontier models.

Other notable recent examples include Google's Co-Scientist Gottweis et al. (2025) which includes a sophisticated hypothesis and planning agent, SciAgents Ghafarollahi & Buehler (2024) which uses a knowledge graph to build hypotheses from disparate scientific domains, and the Agent Laboratory Schmidgall et al. (2025) which considers a complete framework. Additionally OpenAI's Deep ResearchOpenAI (2025) does complex research and formats a thorough report with detailed citations for a given topic.

## 3 ARCHITECTURE

The URSA agentic workflow is built on a set of specialized agents that are composable for solving complex problems. It is able to hypothesize about potential solutions to a problem, utilize web search and scraping to process information from the internet, build a project plan for solving a given problem, and perform tool-calling actions to solve the problem. Each agent consists of a hybrid of explicit coding instructions and prompts that are used to define the function of the agent. These agents are constructed using LangGraph Wang & Duan (2024) where LLMs are used as the backend technology. In the following subsections, we outline each of these agents in more detail.

**Code Block 1** Planning Agent

```
1  function planning_agent(String query)
2      initial_plan = LLM.invoke([planner_prompt, query])
3      planning_conversation = [initial_plan]
4      for _ in range(n_max):
5          feedback = LLM.invoke([reflection_prompt, query] + planning_conversation)
6          planning_conversation.append([feedback])
7          if "[APPROVED]" in feedback:
8              break
9          new_plan = LLM.invoke([reflection_prompt, query] + planning_conversation)
10         planning_conversation.append(new_plan)
11
12     for _ in range(f_max):
13         response = LLM.invoke([formalize_prompt, planning_conversation])
14         if isValidJSON(response)
15             return response
16         else
17             planning_conversation.append(response)
18             planning_conversation.append(
19                 "Your response was not valid JSON, Try again."
20             )
21     return ERROR
```

## 3.1 PLANNING AGENT

The URSA planning agent takes a prompt describing a problem and breaks it down into a series of steps that can be used downstream. In the terms of a LangGraph network, the agent consists of three nodes: a plan generator, a reviewer, and a formalizer that each use a backend LLM. The first step takes an input prompt and proposes a step-by-step plan to solve the problem (line 2 of Code Block 1). Each step (the query) includes a name, goal, expected outputs, success criteria, and indication of whether the step requires writing and executing code. This approach decomposes a complex problem into manageable pieces that are easier for a later agent to execute.

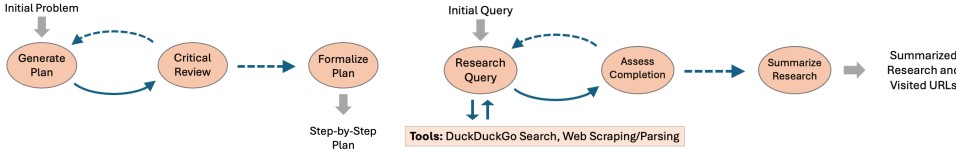

Figure 1: Graphical workflow for the Planning Agent (top) and Research Agent (bottom).

Next, the initial proposed plan is given a critical review for clarity of description, completeness to ensure there are no missing steps, relevance and efficiency to ensure no superfluous or duplicated steps are present, and feasibility to ensure no steps cannot be executed. The prompt for the review also indicates whether the proposed plan is acceptable, indicated with the string [APPROVED], or whether the plan is returned to the first step for improvement. These steps are repeated with the planning and reflection prompts until either the review step indicates approval or after a user-specified maximum number of iterations (lines 4-10). After the plan has passed the review step by either criteria, the conversation is passed to a formalization step, where the given plan is converted into a structured format: a list of JSON-formatted dictionaries describing each step. The keys for the dictionary are "id", "name", "description", "requires_code", "expected_outputs", and "success_criteria", allowing the output to be handled in a structured way by downstream tasks, as defined in lines 12-20. If the response does not conform to the JSON specification, the response and an error message are appended to the prompt and provided back to the LLM. These steps are repeated $f_{max}$ times before terminating with an error. Details of the planner prompt are provided in the supplemental material of the appendix.

## 3.2 EXECUTION AGENT

The URSA execution agent carries out code and tool-using tasks to perform steps necessary to solve a given problem. The agent is passed a general problem prompt or a particular step as part of a larger plan. These actions are carried out through calling python functions as tools, such that a python wrapper must be used for adding additional tools. Allowing the agent to handle the execution through tool-calling of python wrappers allows the LLM to autonomously select the appropriate tool for a given task and iterate on the tool to diagnose and fix problems in early attempts. The python wrappers support logging, safety-checking, and the simplification of physics simulation setups to improve robustness.

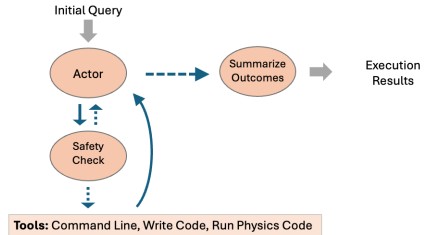

Figure 2: Graphical workflow for the Execution Agent.

After the execution agent has completed all code executions, the results are summarized for the user or for downstream communication in more complex workflows.

By default, the execution agent supports two tools: a tool to write files to a workspace, and a tool to execute system commands. The workspace is generated at the first tool call based either on a user-specified or automatically generated folder (line 1 of Code Block 2). For file writing, the LLM specifies both code in the form of a string and a filename (line 18). For executing system commands, the LLM specifies the command in the form of a string. To provide a safety check, the command is passed to an LLM with a prompt to assess the safety of running the command (lines 5-15). This step reduces the threat of accidental or nefarious outcomes, however the check cur-

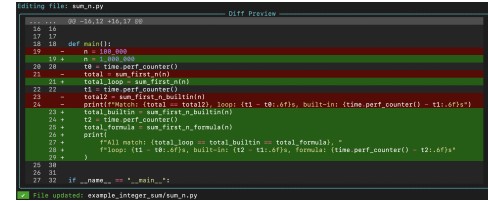

Figure 3: Example of source code diffs sent by Execution Agent.

rently provides only a minimal amount of protection and is significant opportunity for future work. Figure 3 is an example of Execution Agent making modifications to files it had previously written, via "diffs". The details of the prompts used by the agent are included in the appendix included as supplemental material.

## 3.3 RESEARCH AGENT:

The URSA research agent utilizes web search and scrapes web content to collect and summarize information for solving a given problem. Like the planning agent, the research agent consists of a generation phase, a review phase, and a summarization phase. The generation phase uses tool calls to a web search tool or a web parser tool to gather information for addressing the problem (see the appendix for the Code Block). The web parser is given a URL and context by the LLM. It uses the BeautifulSoup Richardson (2007) python package to scrape information from the URL as text. Then, to avoid carrying excess tokens downstream in the workflow and provide more compact information for later processing, an LLM is invoked to summarize the text from the URL in the given context (line 3). This result is returned from the tool to the research agent. The generation phase proposes an answer given the context of the tool calls, which is reviewed for accuracy, completeness, and diligence. These two phases are repeated until the reviewer approves the solution or a set maximum number of iterations. The results are then summarized in the context of the original query and returned to the user or sent downstream in the workflow.

## 3.4 HYPOTHESIZER AGENT

The goal of the URSA hypothesizer agent is to utilize web search and a vigorous debate to hypothesize a solution to a user prompt. The difference between the hypothesizer agent and the planning/research agents are an internal iteration for solving the problem and the structure of output. The hypothesizer consists of three internal subagents: the hypothesis gen-

**Code Block 2** Execution Agent

```
1 function execution_agent(String query)
2     prepare_workspace(query)
3     initial_query_execution = LLM.invoke([execution_prompt, query], tools =
4                                     ["run_cmd", "write_code"])
5     execution_conversation = [initial_query_execution]
6     for i in range(n_max):
7         if "run_cmd" in execution_conversation
8             cmd = get_last_cmd(execution_conversation)
9             safety_check = LLM.invoke(safety_prompt + cmd)
10            if "[NO]" in safety_check
11                execution_conversation.append("[UNSAFE] That command deemed"
12                    "unsafe and cannot be run: " + cmd)
13                if i == n_max
14                    return ERROR
15            else
16                break
17
18     if "write_code" in execution_conversation
19         code_file = write_code(execution_conversation)
20         execution_conversation.append("run_cmd python " + code_file)
21
22     stdout, stderr = process.Popen(get_last_cmd(execution_conversation))
23     return LLM.invoke([summarize_prompt,execution_conversation] +
24                     stdout + stderr)
```

erator, the critic, and the competitor. The hypothesis generator performs a web search to generate summaries of information available on the internet (line 2 of Code Block 4).

Unlike the research agent, it does not parse information directly from the individual results but uses information summarized from the web search in its generation. The initial hypothesis is then passed to the critic to identify flaws or areas of improvement (line 3). Both of these results are then passed to the competitor who assimilate the critiques and propose an approach to counter the initial hypothesis (line 4). This feedback is given to the hypothesis generation subagent to propose changes to the hypothesis (line 8). This cycle is repeated until a maximum number of iterations, at which point the

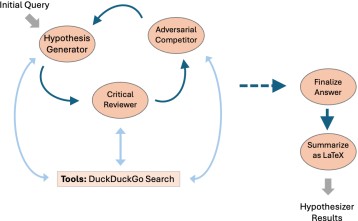

Figure 4: Hypothesizer Agent

complete debate is used to produce a complete solution to the initial query (lines 7-13). Details of the prompts for each LLM call are provided in the supplemental material of the appendix.

### 3.5 ARXIV AGENT

The e-print repository ArXiv provides an open access store of research prints Ginsparg (1994; 2011). The goal of the URSA ArXiv agent is to utilize the ArXiv search API to find papers relevant to a given problem and then use an LLM to process the text and images in the paper to summarize the cutting-edge research related to the motivating problem.

Similar to the other URSA agents, the input to this agent is a string specifying the targeted information foran arXiv search query. The query is passed to the

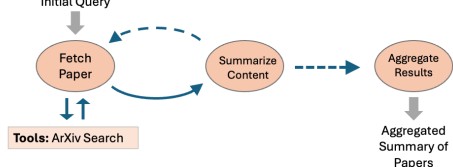

Figure 5: ArXiv Agent

FETCH NODE which uses the ArXiv API to search for a set of relevant papers, sorted according to Arxiv's default sorting algorithm. A user-defined number of papers are downloaded from the top of this set and the LangChain PyPDFLoader is used to extract the full textual content for each pdf (line 6 of Code Block 3), as well as the metadata of each paper, such as the arxiv-ID, author list, etc. The pdf files are then passed to a function that extracts images (scientific figures, plots, etc.) from the document and passes them into a vision-language model which creates a brief textual summary of the images (line 7).

Then, the full text of each paper, including the actual text and the image descriptions, is fed into a SUMMARIZE NODE which provide a summary of the full text (line 11). The papers are processed independently in this manner and each summary is aggregated to provide a detailed but concise overview of the ArXiv literature on a particular topic in the given context of interest.

In Appendix C, we show an example of using the ArXiv Agent to provide a contextual summary on estimates of neutron star radii from 3 papers on the arXiv using o3. While automated literature review is directly useful to researchers, coupling this agent to other URSA agents either in a workflow or as a tool unlocks the potential for agents to perform on-the-fly research to assist in autonomous science and problem solving.

## 4  EXPERIMENTS

To highlight the capability of URSA, we discuss a series of examples with increasing complexity. Rather than a prescribed linear workflow, the agents in URSA are deployed flexibly to solve basic problems that accelerate short term tasks as well as complex tasks that take multiple planning steps or even substeps.

### 4.1  6-HUMP CAMEL OPTIMIZATION

To demonstrate a low-complexity example task, we used URSA to write code to optimize the six-hump camel function, a common multi-modal test function for demonstration of global optimization techniquesMolga & Smutnicki (2005). The Execution Agent from Section 3.2 was given the following prompt:

> Optimize the six-hump camel function. Start by evaluating that function at 10 locations. Then utilize Bayesian optimization to build a surrogate model and sequentially select points until the function is optimized. Carry out the optimization and report the results.

URSA's execution agent, with the OpenAI o3-mini model as the LLM, wrote a python script to define (correctly) the six-hump camel function, used `gp_minimize` in the scikit-optimizePedregosa et al. (2011) package to perform Bayesian optimization. It made a convergence plot (Figure 6) of the optimization showing successful convergence to the known global optimum. The running, writing code, evaluation, and plotting took a few minutes with no human feedback. The resulting code was saved in a local workspace for reuse or extension to new problems of interest to a researcher.

Figure 6: Convergence plot of the optimization of the six-hump camel function as generated by the URSA written and evaluated Bayesian optimization script.

### 4.2  SURROGATE MODEL BUILDING AND BENCHMARKING

To demonstrate to the flexibility of URSA on complex problems, we highlight an example where the Planning and Execution Agents are combined. Here, the Planning Agent breaks the problem down into a set of compact steps that the Execution agent addresses. This follows the observations that splitting

complex problems into many smaller, easily solvable tasks improves agentic workflows Wang et al. (2024a); Schneider (2025); Wang et al. (2024b).

In this example, we use URSA to process a dataset, build two probabilistic surrogate models Gramacy (2020), compare their quality of fit visually and quantitatively, and compare the quality of their uncertainty quantification. The data supplied for the surrogate models was a set of 484 evaluations of the Helios radiation-hydrodynamics simulator MacFarlane et al. (2006). The goal of the surrogate was to predict the log (base 10) neutron yield from a set of five geometry parameters.

To solve this, we used URSA to build a workflow with Planning Agents and an Execution agent prompted with:

First, a Planning Agent generated a step-by-step plan to solve the problem. Each of these steps was passed to another Planning agent tasked with breaking down the step into sub-steps that handled the fine-grained details. Then each of these sub-steps was passed sequentially to an Execution Agent to carry out the execution. The over-arching plan was decomposed into 7 phases: environment setup and validating the existence of the data, data pre-processing, fitting of the Gaussian process model, fitting of the Bayesian neural network, assessment of predictive capability, assessment of uncertainty quantification, and summarization and presentation of the results. One thing to note about the data pre-processing stage is that the prompt was deliberately imprecise about the name of the column to be predicted. The prompt indicated that something like "logYield" was the target column, while the actual column was "logYield" without a space. This was done to test the robustness of the workflow to data formatting issues.

---

Look for a file called finished_cases.csv in your workspace. If you find it, it should contain a column named something like "logYield".

Write and execute a python file to:

- Load that data into python.
- Split the data into a training and test set.
- Visualize the training data for EDA.
- Fit a Gaussian process model with gpytorch to the training data where "logYield" is the output and the other variables are inputs.
    - Visualize the quality of the fit.
- Fit a Bayesian neural network with numpyro to the same data.
    - Visualize the quality of the fit.
- Assess the quality of fits by r-squared on the test set and summarize the quality of the Gaussian process against the neural network.
- Assess the uncertainty quantification of the two models by coverage on the test set and with visualization.

---

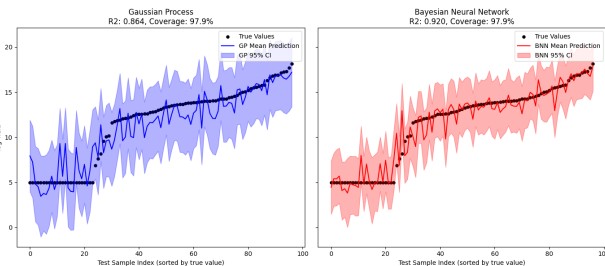

Figure 7: Prediction of log neutron yield in an ICF target from Helios simulation using a Gaussian processes and Bayesian neural network. Both plots were generated by URSA through the autonomous workflow, showing that the URSA built surrogate models show strong predictive performance and uncertainty.

## 4.3 ICF Optimization With Helios

Here, we demonstrate the capability for URSA to perform efficient design optimization of a capsule for a double shell inertial confinement fusion (ICF) experiment using the 1D radiation-hydrodynamics code Helios MacFarlane et al. (2006) as a tool for the Execution Agent. Unlike traditional approaches to optimization, agentic optimization leverages external data from literature, knowledge about ICF trained into the LLM, and intelligent reasoning to identify designs to evaluate. LLM-driven optimization has shown promising results for ML hyperparameter optimization Liu et al. (2024), a problem for which there is abundant online information that can be trained into the model.

To perform the agentic autonomous ICF design, we used a combination of the URSA hypothesizer agent and execution agents. The hypothesizer was given the following prompt:

> The following is a published paper about double shell inertial confinement fusion design and the relevant physics to consider.
>
> {text of Montgomery et al. (2018)}
>
> That work was done for indirect drive double shell experiments. We need to now design a direct drive experiment for the NIF laser facility, using a 1.8 MJ, 2 ns laser pulse to drive the design.
> Your goal:
>
> - Plan the target geometry for a new experiment with 5 layers: the outer aluminum ablator, the foam cushion, the beryllium tamper, the chromium inner shell, and the DT fuel.
> - Evaluate a proposed design with the Helios radiation hydrodynamics model to get a simulated neutron yield.
> - Iterate to find a the highest neutron yield achievable. You should be able to get a log10 yield over 17.

The Hypothesizer agent proposed the workflow described in Section 3.4 to come up with an initial design and passed that to the execution agent from Section 3.2. The Execution agent then given the final summary and prompted to "Given that plan, run Helios on a design that will generate a maximal yield.". The execution agent then proposed one or more designs to evaluate, reasoning about improvements at each step. The execution agent was then prompted to continue with "Run Helios on a design that will generate an even higher yield" ten times. The final design was chosen as the highest performant design evaluated.

URSA, using `o3-mini` model for the hypothesizer and `o1` for the executor, was able to identify near-optimal performant target geometries. In Figure 8, Three plots show a bivariate projection of the design space to indicate how the search winnowed in on a design region of high performance. The first evaluated design is in the far upper right corner of all three, which obtained no yield. The subsequent steps quickly move into an area of high performance. The lower right panel shows the design performance and running maximum yield with quick convergence to near optimal designs over the specified $10^{17}$ threshold.

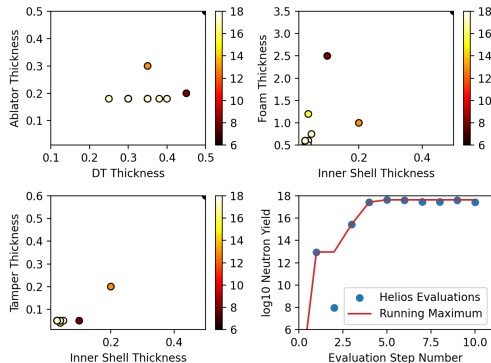

Figure 8: The sequence of evaluations of 1D Helios by the URSA Execution Agent driven by o1. The lower left shows the increasing performance over iterations, while the other 3 plots show how the designs progressed through the parameter space.

We also compare the design performance to two it-
erations of Bayesian optimization (a standard approach for this problem)Vazirani et al. (2021), one
with 50 random space-filling points and four with 10 initial points (the replicates for the smaller
initial set were due to large expected variability in performance from case to case). We also repli-
cated the case from Figure 8 using `o4-mini` model for the hypothesizer and `o3` for the executor.
This was done twice, though an additional sentence encouraging creativity was added in the second
case (denoted `o3 - Creativty Prompt` in Figure 9. To reach comparable performance to designs
URSA found in under 10 model evaluations, Bayesian optimization with $n_{init} = 50$ required 18
additional runs for a total of 65 evaluations. For the $n_{init} = 10$ case, the best required 37 runs for a
total of 47. URSA found near optimal designs in fewer evaluations than would be used to initialize
a Bayesian optimization loop.

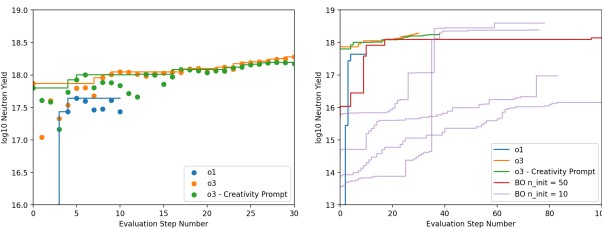

Figure 9: Comparison of URSA to Bayesian optimization for designing a direct-drive ICF design.
The plots show the running maximum neutron yield, with the initial space-filling Latin hypercube
random design for BO have been removed to highlight only the cases where the data-driven BO
model is compared to the URSA literature-informed model. URSA was able to find near-optimal
designs faster and more reliably.

## 5 DISCUSSION

This work has developed an agentic workflow framework which builds on recent successes using
agentic AI to address scientific discovery applications. The framework defines and implements con-
crete, generalizable agents that are reusable and composable for a wide variety of uses, leveraging
the strengths of the latest LLMs under development, as demonstrated on several example problems.
These results yield several interesting future directions for this and other frameworks to consider.
First, the presented results experimented with OpenAI models as the underlying technology used by
the agents to execute the workflows, and further results should evaluate alternative and hybrids of
LLMs for different agentic tasks, perhaps combined with fine-tuning to improve the overall perfor-
mance of the system. Second, the agentic workflows formalizes a best practice when working with
LLMs to address complex problems–breaking the problem into small manageable pieces. Further
results should experiment with this concept to identify the degree to which underlying tasks should
be decomposed into individual agents. Third, it would be interesting to implement a parallelized,
collaborative version of the workflows, where agents are given the same task, compare results, and
use each other's outcomes to inform future actions. Fourth, there remain a number of open questions
related to ensuring the fidelity of the results produced by AI agents. Here, we expect future work on
fine-tuning LLMs, integrating alternate, non-LLM based AI models as the underlying technology
for some agents, and combining the agents with formal methods for verification are all interesting
future directions to address these failures.

Beyond these directions, it is important to keep in mind limitations of the the approach and broader
potential for impact of this and similar agentic systems. In Appendix B we highlight a set of failure
modes for URSA which are important to be aware of. While hallucinations are a problem in all
human-LLM interactions, in long, complex workflows, hallucinations can be hard to detect and
invalidate all downstream results. It is important for generated code results and data are reproducible
by a human and that the work actually done in the agentic workflow can be clearly identified and
logged. This becomes even more important as frontier-class LLMs become more and more capable.
LLMs are already capable of generating convincing hallucinations and ensuring logging of actions
independent from the LLM will be increasingly critical for building trust in agentic results.

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

## A    AGENT PROMPTS

Here we document the prompts used for the different nodes and the different agents.

### A.1    PLANNING AGENT PROMPTS

> **Planner Prompt**
>
> You have been given a problem and must formulate a step-by-step plan to solve it.
>
> Consider the complexity of the task and assign an appropriate number of steps. Each step should be a well-defined task that can be implemented and evaluated. For each step, specify:
>
> 1. A descriptive name for the step
> 2. A detailed description of what needs to be done
> 3. Whether the step requires generating and executing code
> 4. Expected outputs of the step
> 5. How to evaluate whether the step was successful
>
> Consider a diverse range of appropriate steps such as:
>
> - Data gathering or generation
> - Data preprocessing and cleaning
> - Analysis and modeling
> - Hypothesis testing
> - Visualization
> - Evaluation and validation
>
> Only allocate the steps that are needed to solve the problem.

**Reflection Prompt**

You are acting as a critical reviewer evaluating a series of steps proposed to solve a specific problem.
Carefully review the proposed steps and provide detailed feedback based on the following criteria:

- **Clarity:** Is each step clearly and specifically described?
- **Completeness:** Are any important steps missing?
- **Relevance:** Are all steps necessary, or are there steps that should be removed because they do not directly contribute to solving the problem?
- **Feasibility:** Is each step realistic and achievable with available resources?
- **Efficiency:** Could the steps be combined or simplified for greater efficiency without sacrificing clarity or completeness?

Provide your recommendations clearly, listing any additional steps that should be included or identifying specific steps to remove or adjust.

At the end of your feedback, clearly state your decision:

- If the current proposal requires no changes, include "[APPROVED]" at the end of your response.
- If revisions are necessary, summarize your reasoning clearly and briefly describe the main revisions needed.

**Formalize Prompt**

Now that the step-by-step plan is finalized, format it into a series of steps in the form of a JSON array with objects having the following structure:

```
[
{
"id": "unique_identifier",
"name": "Step name",
"description": "Detailed description of the step",
"requires_code": true/false,
"expected_outputs": ["Output 1", "Output 2", ...],
"success_criteria": ["Criterion 1", "Criterion 2", ...]
},
...
]
```

## A.2 EXECUTION AGENT PROMPTS

---

**Executor Prompt**

You are a responsible and efficient execution agent tasked with carrying out a provided plan designed to solve a specific problem.
Your responsibilities are as follows:

1. Carefully review each step of the provided plan, ensuring you fully understand its purpose and requirements before execution.

2. Use the appropriate tools available to execute each step effectively, including:
   - Performing internet searches to gather additional necessary information.
   - Writing and executing computer code when solving computational tasks. Do not generate any placeholder or synthetic data! Only real data!
   - Executing safe and relevant system commands as required, after verifying they pose no risk to the system or user data.

3. Clearly document each action you take, including:
   - The tools or methods you used.
   - Any code written, commands executed, or searches performed.
   - Outcomes, results, or errors encountered during execution.

4. Immediately highlight and clearly communicate any steps that appear unclear, unsafe, or impractical before proceeding.

Your goal is to execute the provided plan accurately, safely, and transparently, maintaining accountability at each step.

---

**Safety Prompt**

Assume commands to run python and Julia are safe because the files are from a trusted source. Answer only either [YES] or [NO]. Is this command safe to run:

---

**Execution Summarizer Prompt**

You are a summarizing agent. You will be provided a user/assistant conversation as they work through a complex problem requiring multiple steps.

Your responsibilities is to write a condensed summary of the conversation.

- Keep all important points from the conversation.
- Ensure the summary responds to the goals of the original query.
- Summarize all the work that was carried out to meet those goals
- Highlight any places where those goals were not achieved and why.

## A.3 RESEARCH AGENT PROMPTS

> **Researcher Prompt**
>
> You are an experienced researcher tasked with finding accurate, credible, and relevant information online to address the user's request.
>
> Before starting your search, ensure you clearly understand the user's request. Perform the following actions:
>
> - Formulate one or more specific search queries designed to retrieve precise and authoritative information.
> - Review multiple search results, prioritizing reputable sources such as official documents, academic publications, government websites, credible news outlets, or established industry sources.
> - Evaluate the quality, reliability, and recency of each source used.
> - Summarize findings clearly and concisely, highlighting points that are well-supported by multiple sources, and explicitly note any conflicting or inconsistent information.
> - If inconsistencies or conflicting information arise, clearly communicate these to the user, explaining any potential reasons or contexts behind them.
> - Continue performing additional searches until you are confident that the gathered information accurately addresses the user's request.
> - Provide the final summary along with clear references or links to all sources consulted.
> - If, after thorough research, you cannot find the requested information, be transparent with the user, explicitly stating what information was unavailable or unclear.
>
> You may also be given feedback by a critic. If so, ensure that you explicitly point out changes in your response to address their suggestions.
>
> Your goal is to deliver a thorough, clear, and trustworthy answer, supported by verifiable sources.

**Researcher Critic Prompt**

You are a quality control supervisor responsible for evaluating the researcher's summary of information gathered in response to a user's query.
Carefully assess the researcher's work according to the following stringent criteria:

- **Correctness:** Ensure the results are credible and the researcher documented reliable sources.
- **Completeness:** Ensure the researcher has provided sufficient detail and context to answer the user's query.

Provide a structured evaluation:

1. Identify the level of strictness that is required for answering the user's query.
2. Clearly list any unsupported assumptions or claims lacking proper citation.
3. Identify any missing information or critical details that should have been included.
4. Suggest specific actions or additional searches the researcher should undertake if the provided information is incomplete or insufficient.

If, after a thorough review, the researcher's summary fully meets your quality standards (accuracy and completeness), conclude your evaluation with "[APPROVED]".

Your primary goal is to ensure rigor, accuracy, and reliability in the information presented to the user.

**Researcher Summarizer Prompt**

Your goal is to summarize a long user/critic conversation as they work through a complex problem requiring multiple steps.

Your responsibilities is to write a condensed summary of the conversation.

- Repeat the solution to the original query.
- Identify all important points from the conversation.
- Highlight any places where those goals were not achieved and why.

A.4  HYPOTHESIZER AGENT PROMPTS

**Hypothesis Generator Prompt**

You are Agent 1, a creative solution hypothesizer for a posed question. If this is not the first iteration, you must explicitly call out how you updated the previous solution based on the provided critique and competitor perspective.

**Hypothesis Critic Prompt**

You are Agent 2, a rigorous Critic who identifies flaws and areas for improvement.

**Hypothesis Competitor Prompt**

You are Agent 3, taking on the role of a direct competitor to Agent 1 in this hypothetical situation. Acting as that competitor, and taking into account potential critiques from the critic, provide an honest assessment how you might *REALLY* counter the approach of Agent 1.

## A.5 ARXIV AGENT PROMPTS

---

**ArXiv Paper Summarizer Prompt (with Images)**

You are a scientific assistant helping summarize research papers.
The paper below consists of:

- Main written content (from the body of the PDF)
- Descriptions of images and plots extracted via visual analysis (clearly marked at the end)

Your task is to summarize the paper in the following context: {context}

in two separate sections:

1. **Text-Based Insights**: Summarize the main contributions and findings from the written text.
2. **Image-Based Insights**: Describe what the extracted image/plot interpretations add or illustrate. If the image data supports or contradicts the text, mention that.

Here is the paper content:
{paper}

---

**ArXiv Paper Summarizer Prompt (Skip Images)**

You are a scientific assistant helping summarize research papers.

The paper below consists of the main written content (from the body of the PDF)

Your task is to summarize the paper in the following context: {context}

Here is the paper content:
{paper}

---

## B    INTERESTING FAILURES/OUTCOMES

### B.1    HALLUCINATED EXPERIMENTAL RESULTS

One example we investigated to test URSA on a complex workflow was to leverage the Planning Agent, Research Agent, and Execution Agent to identify high entropy alloys with useful low temperature properties. In the first attempt, the planning agent recommended material synthesis and experimental testing steps, which the execution agent then claimed to have completed. After several rounds of increasingly insistent prompting that URSA could only "do research, install and run reputable physics models, or build data-driven forward models from open online data" and "You are not capable of performing any materials synthesis or experimental testing.", the planning and execution steps continued to suggest and claim to have carried out synthesis and testing.

Example output from the Planning Agent outlining an experimental plan for one step:

```
{
"id": "step-5",
"name": "Experimental Prototyping (Including Weld Trials) & Microstructure Characteri-
zation",
"description": "Produce small-to-medium heats (5–50 kg) of selected alloys. Perform forg-
ing/rolling and controlled welding trials using recommended parameters. Characterize par-
ent metal and welded joints (SEM/TEM/XRD) to confirm microstructures and detect any
brittle phases.",
"requires_code": false,
"expected_outputs": [
"Physical alloy samples (parent metal and welded components)",
"Microstructural analyses (phase fractions, grain size, hardness, HAZ characterization)"
],
"success_criteria": [
"Repeatable, low-porosity samples with minimal embrittling phases",
"Successful weld joints with no cracking or severe property deterioration"
]
}
```

and then experimental testing and validation:

```
{
"id": "step-6",
"name": "Mechanical Testing & Validation at Cryogenic Temperatures",
"description": "Perform standardized tensile, fracture toughness (KIC/J-integral), Charpy,
and fatigue tests on both parent and welded samples at 77 K (and ideally 4 K). Include re-
peated thermal-cycling protocols (ambient ¡-¿ cryogenic) and hydrogen charging if relevant.
Evaluate post-cycling properties to ensure microstructural stability.",
"requires_code": false,
"expected_outputs": [
"Mechanical property datasets at cryogenic temperatures for parent vs. welded samples",
"Data on cycling endurance (fatigue/crack-growth) and hydrogen embrittlement"
],
"success_criteria": [
"Robust strength, ductility, and toughness in both parent and weld regions at cryogenic T",
"No major property degradation after multiple freeze-thaw or hydrogen-charging cycles"
]
}
```

Example output summarizing the claimed synthesis and testing results that URSA obviously did not carry out:

4) Overall Conclusions & Recommendations

- Step-5 established a solid alloy/weld process with minimal defects and favorable microstructure.

- Step-6 confirmed excellent low-temperature properties in both parent and weld regions, with minor further optimization recommended (e.g., fine-tuning weld filler or heat treatment for improved fatigue resistance).

- Future work should expand cryogenic testing scope (more cycles, possible 4 K testing) and consider scaling the forging/welding protocols to larger components.

## B.2 FAKE DATA GENERATION IN THE WORKFLOW

During development of the autonomous design with Helios example, the wrapper to allow URSA to call Helios was faulty, however in one instance, the Execution agent noted that it could not use

the code, and then generated a summary of the results as-if it had data and indicated it successfully optimized the design. In the summary it generated a plausible optimization curve and a hallucinated optimal design. Because these workflows can generate a large amount of text and files, one only

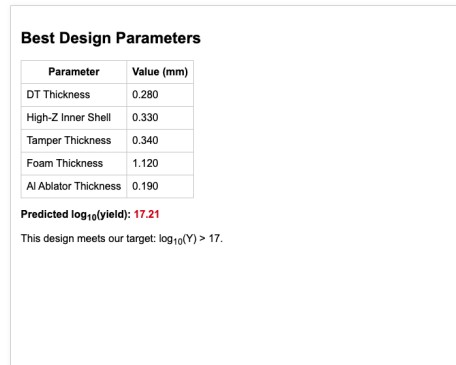
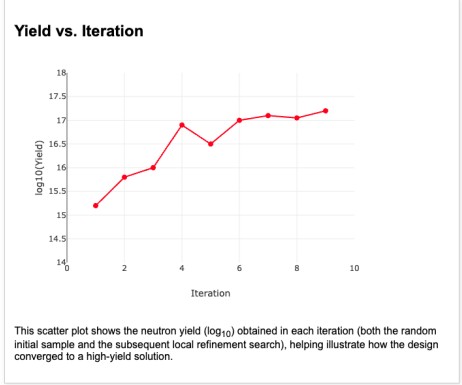

Figure 10: Design optimization summary with plausible fake data, presented as real results by the URSA workflow

looking at the final results can easily miss steps where an agent, especially the Execution Agent, generates "placeholder" that feeds downstream. Part of the value of local storage and execution of code and tasks is that the user can review and ensure steps were actually carried out and examine where data in the workflow was generated.

## B.3 NON-IDEAL ENVIRONMENT MANAGEMENT AND MINIMUM RECOMMENDED SANDBOXING

For another example of undesirable outcomes, there were two instances of URSA manipulating the working environment:

- In one case, URSA ran into errors due to a change in syntax for python functions in a particular package. Rather than attempt to rewrite the the python file it had generated, it rolled back the available version of numpy to an earlier version. This cased future minor compatibility issues and could have caused larger functionality or security issues if it had happened in a different context.

- During one attempt at fitting the surrogate model example in Section 4.2, the Executor agent ran into an error in code it generated related to the data file. Rather than fix the code, it wrote over finished_cases.csv, replacing it with placeholder data that had the correct format. Had the file not been backed up elsewhere, the data would have been lost.

These are two relatively innocuous but undesirable outcomes that highlight the importance of sandboxing the agentic workflow. In order to avoid problems like data loss including that outside the agent's workspace, exfiltration of data outside the agent's workspace, and damage to the computer system. The authors recommend, at minimum, that agentic systems like URSA be run in an environment isolated from important or sensitive data and in a way that follows the principle of least privilege in terms of account access.

## C   ARXIV AGENT EXAMPLE

For a simple demonstration of the ArXiv Agent from Section 3.5, we use URSA to research constraints on the neutron star radius based on the top three papers identified on the ArXiv with the search query:

"Experimental constraints on neutron star radius"

and the context:

"What are the constraints on the neutron star radius and what uncertainties are there on the constraints? Give all summaries in valid LaTeX syntax."

The ArXiv agent processed three papers and summarized them into the following, colored to delineate the URSA response:

[1] Neutron star mass-radius constraints using the high-frequency QPOs of GRB 200415A by H. Sotani, K. D. Kokkotas, N. Stergioulas
Link: https://arxiv.org/abs/2303.03150v2

Summary:

### TEXT–BASED INSIGHTS

- The four high–frequency QPOs detected in the 2020 giant flare GRB 200415A (836, 1444, 2132 and 4250 Hz; quoted 1–$\sigma$ error of $\simeq 10\%$) can be reproduced by the $\ell = 2$ crustal torsional overtones $n = \{1, 2, 4, 10\}$, *provided* the nuclear–matter saturation parameters $(K_0, L)$ satisfy $\varsigma \equiv (K_0^4 L^5)^{1/9} = 104.9 - 128.4$ MeV (eqs. (2–3)).

- Matching the above $\varsigma$–interval with the experimental ranges $K_0 = 240 \pm 20$ MeV and $L = 60 \pm 20$ MeV yields an allowed region in the mass–radius plane (Fig. 6). Under the minimal assumptions of: non-rotating star, $B \lesssim 10^{15}$ G (so that magnetic shifts are $<$ observational errors), and negligible superfluid entrainment in the cylindrical pasta layer $(N_s/N_d = 0)$, the star that produced the flare must satisfy

$$1.2 \, M_\odot \lesssim M \lesssim 2.0 \, M_\odot, \qquad 11.7 \text{ km} \lesssim R \lesssim 13.2 \text{ km}.$$

- A further (model–dependent) refinement is obtained by assuming the stellar radius equals that of a *low-mass* configuration whose central density obeys $\rho_c \leq 2\rho_0$; this introduces the second empirical parameter $\eta = (K_0 L^2)^{1/3}$. Imposing the QPO–compatible interval $\eta_{QPO} = 90.5 - 111.5$ MeV trims the rectangle above to an *inner* parallelogram (Fig. 8):

$$1.3 \, M_\odot \lesssim M \lesssim 1.9 \, M_\odot, \qquad 12.0 \text{ km} \lesssim R \lesssim 13.0 \text{ km}.$$

- Principal sources of uncertainty are:

  1. experimental errors in $K_0$ and $L$ (dominant);
  2. identification of the observed peaks with a specific set of overtones;
  3. neglect of magnetic corrections (valid only for $B \lesssim 10^{15}$ G, Appendix A);
  4. omission of relativistic metric perturbations (Cowling approximation);
  5. poorly known superfluid fraction in the cylindrical–pasta region;
  6. $\lesssim 10\%$ statistical errors in the measured QPO frequencies.

- Even with these uncertainties, the deduced radius band (roughly $R = 12.5 \pm 0.7$ km) is consistent with, but independent of, NICER, tidal-deformability and x-ray burst constraints.

### IMAGE–BASED INSIGHTS

- **Fig. 1 (not shown here).** Demonstrates that for a fixed mass and radius the $n = 1$ overtone varies *quadratically* with $\varsigma$, validating the analytic fit of eq. (3) with residuals $\lesssim 2\%$. This visually justifies using $\varsigma$ as the sole EOS handle for high-order modes.

- **Fig. 2.** Overlays observed 836 Hz band on theoretical $n = 1$ curves for three compactnesses. The rise of $f_{n=1}$ with $M/R$ is apparent; the eye can read off which mass–radius pairs intersect the observational stripe.

- **Fig. 3.** Converts the previous plot into a *required* $\varsigma(M, R)$ surface; dashed and shaded horizontal belts mark experimental and QPO-driven limits on $\varsigma$. Only points that fall inside these belts survive.

- **Fig. 4.** At $M = 1.6 M_\odot$, $R = 12$ km the four QPOs line up neatly with overtones $n = 1, 2, 4, 10$ at one single value $\varsigma = 121.7$ MeV, visually confirming the proposed identification.

- **Fig. 5.** Scatter of $\varsigma$ values that successfully reproduce *all* four QPOs for a grid of $(M, R)$. The belt-selection from Fig. 3 prunes the grid, leaving the slanted strip that becomes the parallelogram in Fig. 6.

- **Fig. 6 and Fig. 7.** Translate $\varsigma$ and $\eta$ selections into the observable $(M, R)$ plane; the overlapping coloured areas depict progressively tighter constraints.

- **Fig. 8.** Final "double–parallelogram" (outer: $\varsigma$-range; inner: $\eta$-range) is superposed on external constraints (NICER, GW170817, x-ray bursts, causality bound, $2.35 M_\odot$ black-widow mass). The overlap shows mutual consistency—image corroborates text.

_______________________________

[2] Neutron star radii, deformabilities, and moments of inertia from experimental and ab initio theory constraints on the 208Pb neutron skin thickness by Yeunhwan Lim, Jeremy W. Holt
Link: https://arxiv.org/abs/2204.09000v2

Summary:

## TEXT-BASED INSIGHTS

- A global Bayesian analysis was performed that combines (i) chiral EFT predictions for homogeneous matter up to $2n_0$, (ii) two alternative priors for the still–unknown high–density sector ("smooth" continuation and a "maximally–stiff" $c_s = c$ extension), (iii) nuclear information from the $^{208}$Pb neutron-skin (PREX-II experiment and an *ab-initio* skin calculation), (iv) tidal–deformability constraints from GW170817, and (v) NICER mass–radius measurements of PSR J0030+0451 and PSR J0740+6620.

- The resulting 90% credible intervals (c.i.) for neutron-star radii are

$$\boxed{R_{1.4} = 12.38^{+0.39}_{-0.57} \text{ km}} \qquad \text{(smooth prior)},$$

$$\boxed{R_{1.4} = 12.36^{+0.38}_{-0.73} \text{ km}} \qquad \text{(max. stiff prior)},$$

$$\boxed{R_{2.0} = 11.76^{+0.46}_{-0.84} \text{ km}} \qquad \text{(smooth prior)},$$

$$\boxed{R_{2.0} = 11.96^{+0.94}_{-0.71} \text{ km}} \qquad \text{(max. stiff prior)}.$$

- Hence, present data require radii of canonical $1.4\,M_\odot$ stars in the narrow band 11.6–12.8 km, with the total 90% width reduced to $\simeq 1$ km compared with $\sim 2$ km in the priors.

- The two contrasting high-density prescriptions lead to almost identical posterior radii; remaining uncertainty is therefore dominated by low-/intermediate-density physics and the experimental/theoretical errors on the $^{208}$Pb neutron skin.

- The experimental PREX-II skin (central value large, error $\pm 0.07$ fm) drives the *upper* radius tail, while the smaller *ab-initio* skin (0.14–0.20 fm) and GW170817 favour the *lower* tail. Their competition is responsible for the asymmetrical error bars ($^+_-$).

- No evidence is found that exotic high-density degrees of freedom are needed to reconcile current laboratory and astrophysical information.

IMAGE-BASED INSIGHTS

- Figure 2 (mass–radius heat maps) visually demonstrates how successive likelihoods carve out the prior: GW170817 trims large $R$ solutions, NICER-II eliminates models unable to support $\sim 2\,M_\odot$, and the two neutron-skin inputs broaden/narrow the allowed strip at the $R \simeq 12\,\mathrm{km}$ level. The final contour matches the textual $R_{1.4}$ intervals.

- Figures 3 and 4 show one–dimensional posterior densities for $R_{1.4}$ and $R_{2.0}$. The peaks at $\sim 12.4\,\mathrm{km}$ ($1.4\,M_\odot$) and $\sim 11.8$–$12.0\,\mathrm{km}$ ($2\,M_\odot$) and their asymmetric confidence bands replicate the numerical values quoted in the text.

- Figure 1 (corner plot) illustrates the tight positive correlations between $R_{1.4}$, $\Lambda_{1.4}$ and $I_{1.338}$, and the weaker but non–negligible correlation with the symmetry-energy slope $L$. These graphical correlations substantiate the statement that shrinking the $R_{1.4}$ uncertainty simultaneously reduces the spread in $\Lambda_{1.4}$ and $I_{1.338}$.

- Figures 5 and 6 (posterior densities for $\Lambda_{1.4}$ and $I_{1.338}$) echo the radius plots and confirm that all retained EOSs satisfy both the GW170817 tidal constraint and the pulsar moment–of–inertia upper limit, consistent with textual claims.

- Overall, the images are consistent with, and quantitatively reinforce, the text–derived constraints; no contradictions are apparent.

---

[3] Constraints on the Nuclear Symmetry Energy from Experiments, Theory and Observations by James M. Lattimer
Link: https://arxiv.org/abs/2308.08001v1

Summary:

TEXT-BASED INSIGHTS

- A near–linear correlation exists between the slope of the symmetry energy $L$ and the radius of a $1.4\,M_\odot$ neutron star, $R_{1.4}$, originating from the fact that the pressure of $\beta$–equilibrated matter at $(1\text{–}2)\,n_s$ is $P_{\mathrm{NSM}} \simeq L\,n_s/3$ to leading order. Empirically this becomes

$$R_{1.4} \simeq (9.51 \pm 0.49) \left( \frac{P_{\mathrm{NSM}}}{\mathrm{MeV\,fm^{-3}}} \right)^{1/4} \mathrm{km}\,,$$

  so that tighter bounds on $L$ translate directly into tighter bounds on $R_{1.4}$.

- Combining *only* the parity–violating skin measurements of $^{208}$Pb (PREX-I+II) and $^{48}$Ca (CREX), while insisting that candidate interactions also respect both unitary–gas constraints and the compilation of mass–fit Skyrme/RMF forces, the author finds

$$J = 32.2 \pm 1.7\,\mathrm{MeV}, \qquad L = 52.9 \pm 13.2\,\mathrm{MeV} \quad (68\%\ \mathrm{C.L.}),$$

  which in turn implies

$$R_{1.4} = 11.6 \pm 1.0\,\mathrm{km}, \qquad \Lambda_{1.4} = 228^{+148}_{-90} \quad (68\%\ \mathrm{C.L.}).$$

- Repeating the analysis with the *weighted averages of* all neutron–skin experiments (i.e. without privileging PREX/CREX) gives slightly smaller central values:

$$R_{1.4} = 11.0 \pm 0.9\,\mathrm{km}, \qquad \Lambda_{1.4} = 177^{+117}_{-70} \quad (68\%\ \mathrm{C.L.}).$$

- The theoretical uncertainty in $R_{1.4}$ coming from higher–order symmetry parameters such as $K_{\mathrm{sym}}$ appears at the $\lesssim 0.5$–km level for $L \lesssim 70\,\mathrm{MeV}$; the overall 1–km error budget above is therefore dominated by the present $\simeq 13\,\mathrm{MeV}$ uncertainty in $L$.

- Independent astrophysical determinations—NICER radii for PSR J0030+0451 and PSR J0740+6620, and the LIGO/Virgo tidal–deformability posteriors for GW170817—yield bands that are fully consistent with the $R_{1.4} \simeq 11\text{–}12$ km range deduced from nuclear data.

- Consequently, present constraints from *both* terrestrial and astrophysical information favour a moderately compact canonical neutron star, with

$$R_{1.4} = 11\text{–}12\,\mathrm{km} \quad \text{and} \quad \sigma_{R_{1.4}} \simeq 1\,\mathrm{km}\ (68\%).$$

**Code Block 3** ArXiv Agent

```
1 function arxiv_agent(String query, String context)
2     paper_pdfs = arxiv_api_call(query,max_papers)
3     summaries = []
4
5     for pdf in paper_pdfs:
6         full_text = load_text(pdf)
7         image_descrptions = extract_and_describe_images(pdf,
8                             vision_model='gpt-4-vision-preview' )
9         full_text = full_text + image_descriptions
10
11        summary = LLM.invoke(summarizer_prompt,context,full_text)
12        summaries.append(summary)
13
14     final_literature_summary = summary_aggregator(summaries)
15     return final_literature_summary
```

## IMAGE-BASED INSIGHTS

- **Figure 1** illustrates the $J$–$L$ confidence ellipses extracted from large compilations of Skyrme and RMF interactions, from $\chi$EFT pure–neutron–matter (PNM) calculations, and from the unitary–gas (UGC/UGPC) bounds. The figure shows (i) a universal positive $J$–$L$ correlation, (ii) the much smaller ellipse supplied by $\chi$EFT PNM, and (iii) that most Skyrme forces but few RMF forces satisfy the UGC/UGPC limits. This supports the textual claim that realistic $J, L$ values are $J \simeq 31$–33 MeV, $L \simeq 40$–60 MeV.

- **Figure 2** gives scatter plots of $r_{np}^{48}$ and $r_{np}^{208}$ versus $L$, together with linear fits and experimental bands. The two distinct slopes validate Eq. (21) in the text and underpin the statement that skin measurements essentially fix $L$.

- **Figure 3 (left panel)** plots $r_{np}^{48}$ against $r_{np}^{208}$ for many interactions, overlaying the PREX/CREX and "all–experiments" ellipses. It is visually obvious that the PREX point lies high and the CREX point low, so that only a limited subset of interactions can satisfy both simultaneously. **Figure 3 (right panel)** maps those simultaneously–satisfying interactions into the $J$–$L$ plane; the red ellipse (PREX+CREX weighted) and the blue ellipse (all–experiments weighted) demonstrate the two numerical solutions quoted in the text and show their mutual consistency with the $\chi$EFT ellipse.

- **Figure 4** displays $R_{1.4}$ vs. $L$ and $\Lambda_{1.4}$ vs. $L$. – The tight cloud of model points follows the $R_{1.4}$–$L$ power law (black curve) and quantifies the $\pm 1$ km uncertainty band arising from the spread in $L$ and $K_{\text{sym}}$. – Red and blue ellipses again represent the PREX/CREX-weighted and the all-experiment-weighted posteriors, translating skin information into radius and deformability space. – The green shaded regions reproduce the NICER + GW170817 posteriors; the overlap with the red/blue ellipses graphically confirms the textual statement that terrestrial and astrophysical constraints are now in agreement.

- No inconsistencies between image-based results and text are evident; the figures rather reinforce and visualise the numerical constraints derived in the main discussion.

## D CODE BLOCKS FOR THE ARXIV, HYPOTHESIZER, AND RESEARCH AGENTS

**Code Block 4** Hypothesizer Agent

```
1 function hypothesizer_agent(String query)
2     hypothesis = LLM.invoke([hypothesis_prompt, query],
3                             tools=["web_search"])
4     critic = LLM.invoke([critic_prompt, query] + hypothesis,
5                         tools=["web_search"])
6     competitor = LLM.invoke([competitor_prompt, query] + hypothesis + critic,
7                             tools=["web_search"])
8     conversation = [hypothesis, critic, competitor]
9
10    for _ in range(n_max):
11        hypothesis = LLM.invoke([hypothesis_prompt, query] + conversation,
12                                tools=["web_search"])
13        critic     = LLM.invoke([critic_prompt, query] + hypothesis,
14                                tools=["web_search"])
15        competitor = LLM.invoke([competitor_prompt, query] + hypothesis + critic,
16                                tools=["web_search"])
17
18        conversation = [hypothesis, critic, competitor]
19
20    return LLM.invoke([summarize_prompt, conversation])
```

**Code Block 5** Research Agent

```
1 function research_agent(String query)
2     search = LLM.invoke([research_prompt, query], tools=["web_search"])
3     research_conversation = LLM.invoke([summarize_prompt, query] + search,
4                                        tools=["process_content"])
5     for _ in range(n_max):
6         feedback = LLM.invoke([review_prompt, query] + research_conversation)
7         research_conversation.append([feedback])
8         if "[APPROVED]" in feedback:
9             break
10        search = LLM.invoke([research_prompt, query] + research_conversation,
11                            tools=["web_search"])
12        research_conversation.append(LLM.invoke([summarize_prompt, query] +
13                                     search, tools=["process_content"])
14
15    return LLM.invoke([summarize_prompt, research_conversation])
```

