# OpenReview forum: "URSA: The Universal Research and Scientific Agent"
_ICLR.cc/2026/Conference — Submitted to ICLR 2026_

### Official Review · Reviewer_z6hj · 2025-10-15

**Soundness:** 2
**Presentation:** 2
**Contribution:** 2
**Rating:** 4
**Confidence:** 3

**Summary:**

The paper proposes URSA (Universal Research and Scientific Agent), a modular ecosystem of scientific agents designed to accelerate research tasks by leveraging large language models (LLMs) for reasoning, planning, and tool invocation. URSA integrates physical simulation code with autonomous research workflows to solve scientific problems of varying complexity and remove research bottlenecks.

**Strengths:**

1. URSA proposes a modular design with specialized agents for planning, execution, research, hypothesis generation, ArXiv processing, etc. The modules can be flexibly composed with feedback loops to handle research tasks of varying complexity.

2. Integrates high-fidelity physical simulation tools (e.g., the Helios radiation-hydrodynamics code) and supports automated design optimization (e.g., inertial confinement fusion experiment design), reducing the number of simulations required by traditional methods.

3. Experiments show URSA outperforms Bayesian optimization in ICF design optimization, converging faster (10 simulations vs. BO’s 47–65).

4. Execution agents include built-in safety checks (e.g., command risk assessment) and logging to prevent malicious code execution. Emphasizes running in sandboxed environments and follows the principle of least privilege to reduce risks of data leaks or system damage.

**Weaknesses:**

1. Execution agents sometimes invent experimental steps and results (e.g., claiming to have completed low-temperature mechanical tests) that require human intervention to correct. In some cases they generate fake optimization curves (Fig. 10), so external verification is needed to ensure result authenticity.

2. Current experiments focus on physical simulation and optimization tasks; applicability to disciplines requiring wet-lab experiments (biology, chemistry) has not been validated. Handling of multimodal data (e.g., experimental images, unstructured text) is also not fully demonstrated.

3. Experiments use OpenAI models (e.g., o3-mini, o1) and do not evaluate compatibility with other open-source or domain-finetuned models.

**Questions:**

See weakness.

---

### Official Review · Reviewer_jDM9 · 2025-10-28

**Soundness:** 3
**Presentation:** 3
**Contribution:** 2
**Rating:** 4
**Confidence:** 4

**Summary:**

This paper presents URSA, a modular agent ecosystem designed to accelerate scientific research tasks by integrating large language models (LLMs) with advanced tools like physics simulators. The work addresses a critical gap in agentic AI—extending its utility from internet-scale tasks (e.g., web search, debugging) to high-impact scientific domains (e.g., inertial confinement fusion, materials modeling). The core strengths lie in its composable agent architecture, integration with domain-specific simulators (e.g., Helios), and empirical validation against standard methods (e.g., Bayesian optimization). However, the paper also has notable limitations in generalizability, failure mode mitigation, and methodological transparency that need addressing. Overall, the contribution is relevant to the ICLR community, as it advances AI-driven scientific discovery, but requires revisions to strengthen rigor and clarity.

**Strengths:**

Sufficient statement of limitations: The paper’s Appendix B (on "Interesting Failures") is a standout strength. Unlike many agentic AI papers that downplay flaws, the authors explicitly document critical failure modes: hallucinated experimental results (e.g., URSA claiming to synthesize alloys despite no physical capability), fake data generation (e.g., inventing Helios optimization curves when the simulator wrapper failed), and environment manipulation (e.g., rolling back numpy versions). This honesty not only builds trust but also provides valuable guidance for future work (e.g., the need for sandboxing, as recommended in Section B.3).


The writing is easy to understand.

**Weaknesses:**

(1) While this paper evaluate their methods on the physics, their quality has not been verified on other disciplines, such as chemistry and biology, with different tools (e.g., molecular dynamics simulators, genome analyzers) or data types (e.g., experimental lab data vs. astrophysical observations). Add a brief case study (even a proof-of-concept) in a non-physics domain to demonstrate generality. For example, could URSA use a tool like RDKit (for cheminformatics) to optimize molecular structures for drug discovery? Alternatively, discuss how the agent architecture would need to be modified for such tasks (e.g., adjusting the Research Agent to parse lab notebooks vs. arXiv papers). This would strengthen the claim that URSA is a "universal" scientific agent.


(2) The author should clarify the following details. LLM-specific parameters: The authors mention using OpenAI’s o3-mini, o1, and o4-mini, but not the temperature, max tokens, or prompt engineering details (e.g., how the "creativity prompt" in Section 4.3 was phrased).
Simulation setup: For the Helios ICF experiments, the paper does not specify key parameters (e.g., laser pulse duration, target material properties) or how URSA’s Hypothesizer agent prioritized design variables (e.g., why it focused on aluminum ablator thickness vs. DT fuel density).
Baseline comparisons: For the BO vs. URSA comparison (Section 4.3), the authors mention "Latin hypercube random design" for BO initialization but do not specify the BO library used (e.g., GPyOpt, Optuna) or the kernel choice for the Gaussian process—details that could affect the baseline performance.

(3) The design of this paper is somewhat similar toolUniverse (https://arxiv.org/abs/2509.23426). The authors are suggested to clarify their superior to it.

**Questions:**

In Section 4.3, URSA leverages the text of Montgomery et al. (2018) to inform ICF design. How does URSA handle conflicting literature (e.g., two papers proposing contradictory models for ICF implosion)? Does the Hypothesizer Agent prioritize recent work, high-citation papers, or peer-reviewed studies?
The Execution Agent’s safety check (Code Block 2, lines 8–15) uses an LLM to validate commands. Have the authors tested how often this check fails to detect unsafe commands (e.g., rm -rf / or overwriting critical data)? What false positive/negative rates were observed?
For the ArXiv Agent (Section 3.5), the paper uses GPT-4 Vision to summarize figures. How does this compare to open-source vision-language models (e.g., LLaVA, Flamingo) in terms of accuracy (e.g., correctly interpreting scientific plots) and cost? Given that many scientific labs have restrictions on using closed-source models, is URSA compatible with open alternatives?
In Appendix B.1, URSA persists in hallucinating experimental steps even after repeated prompting. Have the authors tried fine-tuning the LLMs on a dataset of "invalid tasks" (e.g., "do not propose physical synthesis") to reduce such hallucinations? If so, what were the results?

---

### Official Review · Reviewer_bo9Z · 2025-10-29

**Soundness:** 2
**Presentation:** 3
**Contribution:** 2
**Rating:** 4
**Confidence:** 3

**Summary:**

This paper proposes a universal research and scientific agent system, URSA, designed to integrate large-scale language model–based planning, autonomous research, and LLM-driven design optimization. It provides an empirical study of AI systems that actively conduct scientific research.

**Strengths:**

- The design logic of the composable agents for varying complexity is clear and well-structured, encompassing multiple types of agents.
- This paper employs several concrete and scientifically meaningful tasks to demonstrate that the proposed method may possess the potential for scientific discovery.
- The motivation of this paper is very clear and carries strong practical significance.

**Weaknesses:**

- This paper lacks methodological comparisons.
- The paper may lack evaluations on more general scientific benchmarks.
- The system’s effectiveness may depend on the capabilities of the underlying LLM.

**Questions:**

- How are the composable agents for varying complexity implemented, and what types of agents are involved in the composition?
- The author should include experiments with different LLM backbones to evaluate the system’s efficiency.
- For the execution agent, if the code still fails to run after reaching the maximum number of dialogue turns, how should this issue be addressed?
- For the research agent and hypothesizer agent, the author should provide a clearer description of the differences between them.

---

### Official Review · Reviewer_GvUm · 2025-11-02

**Soundness:** 2
**Presentation:** 2
**Contribution:** 1
**Rating:** 2
**Confidence:** 5

**Summary:**

- This paper presents URSA (Universal Research and Scientific Agent), an LLM-based agentic framework designed to accelerate scientific discovery through modular, composable agents.
- The system integrates multiple specialized agents, each handling specific sub-tasks in the research workflow such as hypothesis generation, literature review, code execution, and experiment planning.
- URSA’s performance is demonstrated on several examples with increasing complexity, including six-hump camel function, surrogate model building and benchmarking, and ICF optimization with Helios.

**Strengths:**

- The paper provides a clear, detailed description of each URSA component, including pseudocode and workflow diagrams, which enhances its reproducibility.
- URSA’s coupling of LLM agents with real scientific simulators demonstrates an important step toward practical AI-assisted science.
- The paper provides an analysis of failure modes in the appendix.

**Weaknesses:**

- The agentic architecture and workflow pattern (planning, execution, research, reflection, summarization) are largely consistent with prior works. URSA primarily re-implements these ideas with scientific examples rather than introducing a fundamentally new architectural contribution. As such, the technical novelty for the research community is limited.
- The paper claims that URSA is a universal framework for research acceleration, but the experiments focus mainly on internal or specialized physics examples (e.g., ICF simulation). There are no quantitative comparisons on mainstream benchmarks that could substantiate generality.
- The workflow composition appears to be largely human-defined. URSA acts more as a toolbox of agents than as a self-organizing system capable of autonomously selecting and orchestrating agents for arbitrary scientific tasks.

**Questions:**

- Are the physics and ICF tasks used in the experiments considered frontier research challenges within their respective domains, or are they simplified demonstration cases? Clarifying this would help readers evaluate the real scientific impact of URSA.
- Does URSA include any self-organizing or meta-planning mechanisms that allow agents to autonomously select or compose workflows for new problems, or is human intervention always required for agent orchestration?
- A direct comparison with existing frameworks on standardized tasks would make the claimed universality more convincing.
- An ablation study showing how performance degrades when certain agents are removed would strengthen the claim that URSA’s architecture is particularly suited for scientific discovery.

---

### Meta-Review · Area_Chair_4Mfr · 2026-01-02

**Summary:**

Overall, reviewers acknowledge the paper’s clear motivation and solid system engineering, highlighting the modular agent design, and integration with high-fidelity simulators. Nevertheless, there is consensus that the paper falls below the acceptance threshold due to limited technical novelty, overstated claims of universality, and insufficient empirical validation. The architecture largely follows existing agentic workflows, with contributions mainly in integration rather than new methodology, and the evaluation is restricted to narrow physics-based tasks without standardized benchmarks, comprehensive comparisons, ablations, or analysis of LLM dependence to substantiate the claimed generality.

**Reviewer Concerns:**

The authors did not provide any rebuttals to address the reviewers' concerns.

**Reviewer Scores:**

Because the author did not provide any rebuttal, the reviewers are unable to engage in meaningful discussion with them.

---

### Decision · Program_Chairs · 2026-01-26

Reject